# A Novel Method of Bridge Deflection Prediction Using Probabilistic Deep Learning and Measured Data

**DOI:** 10.3390/s24216863

**Published:** 2024-10-25

**Authors:** Xinhui Xiao, Zepeng Wang, Haiping Zhang, Yuan Luo, Fanghuai Chen, Yang Deng, Naiwei Lu, Ying Chen

**Affiliations:** 1School of Civil Engineering, Hunan University of Technology, Zhuzhou 412007, China; xiaoxinhui@hut.edu.cn (X.X.); wangzepeng@hut.edu.cn (Z.W.); luoyuan@hut.edu.cn (Y.L.); chengfanghuai@hut.edu.cn (F.C.); 2School of Civil and Transportation Engineering, Beijing University of Civil Engineering and Architecture, Beijing 100044, China; dengyang@bucea.edu.cn; 3School of Civil Engineering, Changsha University of Science and Technology, Changsha 410114, China; lunaiwei@csust.edu.cn; 4Hunan Urban Construction College, Xiangtan 411104, China; chenyin@hut.edu.cn

**Keywords:** bridge deflection, probability neural network, structural health monitoring, interval prediction, suspension bridge, gaussian distribution

## Abstract

The deflection control of the main girder in suspension bridges, as flexible structures, is critically important during their operation. To predict the vertical deflection of existing suspension bridge girders under the combined effects of stochastic traffic loads and environmental temperature, this paper proposes an integrated deflection interval prediction method based on a Convolutional Neural Network (CNN), Long Short-Term Memory (LSTM), a probability density estimation layer, and bridge monitoring data. A time-series training dataset consisting of environmental temperature, vehicle load, and deflection monitoring data was built based on bridge health monitoring data. The CNN-LSTM combined layer is used to capture both local features and long-term dependencies in the time series. A Gaussian distribution (GD) is adopted as the probability density function, and its parameters are estimated using the maximum likelihood method, which outputs the optimal deflection prediction and probability intervals. Furthermore, this paper proposes a method for identifying abnormal deflections of the main girder in existing suspension bridges and establishes warning thresholds. This study indicates that, for short time scales, the CNN-LSTM-GD model achieves a 47.22% improvement in Root Mean Squared Error (RMSE) and a 12.37% increase in the coefficient of determination ($R^{2}$) compared to the LSTM model. When compared to the CNN-LSTM model, it shows an improvement of 28.30% in RMSE and 6.55% in $R^{2}$. For long time scales, the CNN-LSTM-GD model shows a 54.40% improvement in RMSE and a 10.22% increase in $R^{2}$ compared to the LSTM model. Compared to the CNN-LSTM model, it improves RMSE by 38.43% and $R^{2}$ by 5.31%. This model is instrumental in more accurately identifying abnormal deflections and determining deflection thresholds, making it applicable to bridge deflection early-warning systems.

## 1. Introduction

Deflection plays a critical role within a structural health monitoring (SHM) system when the operational condition of a bridge is assessed. It serves as a direct indicator of the bridge structure’s overall vertical stiffness and load-carrying capacity, offering valuable insights into changes in bridge alignment [1,2,3,4]. Throughout the operational lifespan of a bridge, the deflection signals collected by the SHMS primarily arise from the combined effects of vehicle loads, environmental loads, material degradation, ambient noise, and various other factors [5,6]. Notably, vehicle loads and environmental temperature emerge as the principal factors influencing bridge deflection [7,8,9].

Currently, numerous scholars are dedicated to investigating the influence of temperature on deflection monitoring data, with research efforts falling into two main categories. First, methods aimed at isolating the temperature effect from deflection data are widely studied. Commonly employed techniques for separating temperature effects include wavelet analysis, empirical modal decomposition (EMD), and ensemble empirical modal decomposition (EEMD) [10,11,12]. For example, Zhao et al. [13] utilized discrete wavelet variations to distinguish temperature and train components within deflection monitoring data, while Wu et al. [14] introduced ensemble empirical modal decomposition to isolate vehicle-load effects in bridge monitoring, enhancing the performance of empirical modal decomposition. Addressing issues such as EMD modal aliasing and cumulative errors, Li et al. [15] proposed a decomposition method termed time-varying filtered empirical mode decomposition, based on time-varying filtered empirical modal decomposition, permutation entropy, and Kullback–Leibler scattering (TVFEMD-PE-KLD). This method effectively monitors deflection data attributed to temperature-induced effects. The second category of research focuses on establishing the correlation between temperature and temperature-induced deflection. Commonly employed correlation research methods include finite element analysis, linear regression, and neural network modeling [16,17,18]. For instance, Zhou et al. [19] developed a multiple linear superposition model to estimate the impact on mid-span deflection in cable-stayed bridges under varying temperature conditions, utilizing a combination of plane geometry analysis and finite element analysis. Similarly, Xu et al. [20] calculated normalization coefficients for different types of temperature effects using a finite element model. They proposed a multivariate linear model to predict temperature-induced deflection, considering different spans and locations separately. With the emergence of deep learning, CNN has been widely applied in the health monitoring of civil engineering [21,22,23,24,25]. CNN and LSTM have also been studied in predicting temperature-induced deflection [26,27]. These models offer enhanced accuracy compared to linear regression by addressing issues such as single-point input and time lag effects. They are capable of detecting deflection anomalies at the millimeter level. Additionally, Bayesian neural networks demonstrate superior performance in expressing model uncertainty. Wang et al. [28] introduced an improved Bayesian dynamic linear model for predicting temperature-induced deflection, optimizing model parameters using the Expectation Maximization (EM) and Karman Smoothing (KS) algorithms to obtain deflection and corresponding confidence intervals.

Previous research has somewhat investigated the impact of vehicle loads on the data obtained from deflection monitoring [29,30,31]. For instance, Lu et al. [32] introduced a methodology to assess the probability of the first passage of large-span bridges based on measured data, while also analyzing the maximum displacement of such bridges under vehicle loading conditions. In a related investigation focused on railroad bridges, Zhao et al. [13] delved into the deflection caused by temperature and train effects. They examined the impact of temperature-induced girder deformation on train-induced deflection using online monitoring data and a vehicle–bridge dynamics model, establishing an early warning threshold for main girder deflection under the combined influence of temperature and train. Additionally, Deng et al. [33] employed a gated recurrent unit (GRU) neural network to develop a correlation model. This model utilized vehicle influence coefficients (VICs) and ambient temperature as inputs and deflection data as outputs. Leveraging this trained model, they proposed a method to segregate the temperature and vehicle-loaded components within deflection data. While scholars have extensively researched bridge deflection, their focus has predominantly centered on identifying the factors influencing deflection. However, it is crucial to quantitatively assess the uncertainty inherent in bridge structures during operation and maintenance. This is imperative because errors and uncertainties can significantly impact the safety and reliability of the structure [34,35,36].

In general, uncertainty can be categorized into cognitive uncertainty and random chance uncertainty [37,38,39,40]. Cognitive uncertainty arises from a lack of information or cognitive ability regarding objectively existing things or events, such as the absence of deflection monitoring data or insufficient knowledge. This type of uncertainty can be mitigated by acquiring more data and enhancing understanding of the phenomena. However, accurately representing this uncertainty in practice poses challenges [41,42]. On the other hand, random chance uncertainty typically comprises parametric uncertainties inherent in the structure or component itself, such as manufacturing errors and material variability, as well as errors in data acquisition and processing. While these uncertainties can be quantitatively characterized to some extent, they cannot be entirely eliminated [43].

This paper establishes an integrated deflection interval prediction method based on CNN, LSTM, a probability density estimation layer, and bridge monitoring data. The main contributions of this paper are as follows:(1)To address the uncertainties present in deflection monitoring data, a deflection interval prediction model based on the CNN-LSTM-GD model is proposed. This method not only provides deterministic deflection predictions but also quantifies uncertainty, yielding different confidence intervals. It is applicable for studying the complex nonlinear relationships between deflection, vehicle load, and temperature.(2)The performance of the CNN-LSTM-GD model is tested across different time scales, and a comparative study with CNN-LSTM and LSTM models reveals that the CNN-LSTM-GD model exhibits superior prediction capability, particularly for small deflection fluctuations and extreme deflection events.(3)Based on the uncertainty intervals output from the probabilistic model, two early warning mechanisms are devised, establishing deflection warning thresholds. Compared with conventional statistical analysis methods and finite element methods, this approach is more convenient and efficient, allowing for the timely assessment of bridge safety conditions.

## 2. Deflection Prediction Model Based on Probabilistic Deep Learning

### 2.1. Convolutional Neural Network

Convolutional Neural Networks (CNNs) effectively reduce the number of parameters by employing locally connected layers and a weight-sharing structure, thereby enhancing the model’s sparsity. Compared to fully connected neural networks, this architectural feature enables CNNs to better handle high-dimensional data [44]. One-dimensional convolution (1D-CNN) can slide over time-series data to capture local temporal dependencies. The standard 1D-CNN structure consists of an input layer, a convolutional layer, an activation layer, a pooling layer, and a fully connected layer, as shown in Figure 1. In this study, the vehicle load, temperature, and deflection data sequences $x_{t}^{\mathrm{CNN}}$ are first fed into the input layer. Then, a three-layer 1D-CNN is applied to extract local features from the input data. Considering that this process is linear, an activation function is introduced in the activation layer to apply nonlinear transformations to the output of the convolution layers, thereby enhancing the network’s ability to learn more complex features. Finally, a max-pooling layer is then used to perform downsampling, reducing the data size while retaining key feature information, resulting in the output sequence $y_{t}^{\mathrm{CNN}}$. This process effectively reduces the number of model parameters and computational load, improving training speed and generalization performance. The extracted important feature information is subsequently fed into the LSTM layer. The specific calculation is shown as follows [45]:(1)$$y_{t}^{\mathrm{CNN}}=\max\left\{\mathrm{RELU}\left(W_{C}\cdot x_{t}^{\mathrm{CNN}}+b_{C}\right)\right\}$$

In the equation, $x_{t}^{\mathrm{CNN}}$ represents the input information to the CNN at the current time step; $W_{C}$ and $b_{C}$ denote the weight matrix and bias vector of the CNN, respectively; RELU represents the activation function; and max denotes the max pooling operation.

### 2.2. Long Short-Term Memory

The Long Short-Term Memory (LSTM) network represents a type of Recurrent Neural Network (RNN) specifically designed to capture semantic associations within long sequences, effectively mitigating issues such as gradient vanishing or explosion that commonly afflict the classical RNN [46]. Characterized by a more intricate architecture, LSTM comprises four fundamental components: the cell state, forgetting gate, input gate, and output gate. LSTM and CNN structural diagrams are shown in Figure 2. LSTM not only utilizes the feature information at the current time step but also leverages the feature information retained from the previous time step, allowing it to consider both long-term and short-term memories when processing sequential data. In the LSTM unit, the cell state $C_{t}$ serves as the cornerstone for recording and transmitting information. It achieves the retention of important information and the discarding of irrelevant information through updates and controls at each time step. The forget gate $f_{t}$ determines which information needs to be removed from the cell state. It takes the current input $x_{t}$ and the previous hidden state *h*_*t*−1_ as inputs and outputs a value between 0 and 1, indicating the proportion of information to retain. The input gate $i_{t}$ controls which new information is stored in the cell state. It also accepts $x_{t}$ and *h*_*t*−1_ as inputs, generating a value between 0 and 1. The candidate cell state $\tilde{C_{t}}$ generates new candidate information that will be added to the cell state under the control of the input gate. The cell state is updated based on the outputs of the forget gate, input gate, and candidate cell state. The output gate $o_{t}$ controls how much information is read from the cell state. It generates a value between 0 and 1, which is multiplied by the tanh transformation of the cell state to produce the new hidden state $h_{t}$. 

The key formulas of the LSTM neural unit are as follows [24]:(2)$$f_{t}=\sigma \left(W_{f}\cdot \left[h_{t-1},x_{t}\right]+b_{f}\right)$$
(3)$$i_{t}=\sigma \left(W_{i}\cdot \left[h_{t-1},x_{t}\right]+b_{i}\right)$$
(4)$$\tilde{C_{t}}=\tanh\left(W_{C}\cdot \left[h_{t-1},x_{t}\right]+b_{C}\right)$$
(5)$$C_{t}=f_{t}\cdot C_{t-1}+i_{t}\cdot \tilde{C_{t}}$$
(6)$$o_{t}=\sigma \left(W_{o}\cdot \left[h_{t-1},x_{t}\right]+b_{o}\right)$$
(7)$$h_{t}=o_{t}\cdot \tanh\left(C_{t}\right)$$
where *C_t_*_−1_ and $C_{t}$ represent the cell states at the previous and current time steps, respectively. *h_t_*_−1_ and $h_{t}$ denote the cell outputs at the previous and current time steps. The variable $x_{t}$ represents the input information at the current time step. $\tilde{C_{t}}$ is the candidate cell state at the current time step. The variables $f_{t}$, $i_{t}$, and $o_{t}$ denote the forget gate, the input gate, and the output gate states at the current time step. $W_{f}$, $W_{i}$, $W_{c}$, and $W_{o}$ are the weight matrices for the forget gates, input gates, cell states, and output gates, respectively. The function σ(·) represents the sigmoid function, and the tanh(·) function denotes the hyperbolic tangent function.

### 2.3. Interval Prediction Model Based on Probability Density Estimation

To quantify uncertainties in bridge structures, this paper proposes a deflection interval prediction model that integrates CNN, LSTM, and probability density distribution functions. The model employs a combination of the maximum likelihood estimation approach and deep learning gradient optimization to train on the true deflection values. This enables the model to infer the most probable parameter values of the probability density distribution function and derive both prediction intervals and deterministic prediction values under specified coverage probabilities.

This paper utilizes the Gaussian distribution as the probability density function and estimates its parameters through the maximum likelihood method. Denoting the deflection data by y, the likelihood function is expressed as:(8)$${\left(\mu ,\sigma \right)}_{MLE}=\arg\min\left\{\sum_{i=1}^{n}\right\}-\ln f\left(y_{i},\mu ,\sigma \right)$$

The optimal parameters of the Gaussian distribution can be determined by minimizing the above equation. By substituting the probability density function into the equation and taking the logarithm of both sides, the final loss function is obtained:(9)$$f\left(y_{i},\mu ,\sigma \right)={\left(2\pi {\hat{\sigma }}^{2}\right)}^{-\frac{1}{2}}\exp\left(-\frac{{\left(y_{i}-\hat{\mu }\right)}^{2}}{2{\hat{\sigma }}^{2}}\right)$$
(10)$$loss_{l\left(\hat{\mu },\hat{\sigma }\right)}=-\ln\left\{{\left(2\pi {\hat{\sigma }}^{2}\right)}^{-\frac{1}{2}}\exp\left(-\frac{{\left(y_{i}-\hat{\mu }\right)}^{2}}{2{\hat{\sigma }}^{2}}\right)\right\}=\frac{{\left(y_{i}-\hat{\mu }\right)}^{2}}{2{\hat{\sigma }}^{2}}+\ln\hat{\sigma }+\ln\sqrt{2\pi }$$
where $y_{i}$ represents the deflection value of the training data, and $\hat{\mu }$ and $\hat{\sigma }$ are the estimation parameters output from the probability density estimation layer. To achieve optimal likelihood estimation, it is recommended to employ the Adam optimizer in the probabilistic model to minimize the loss function and ascertain the best expectation and standard deviation of the Gaussian distribution.

The probability density estimation layer serves as the output layer in a fully connected manner. To elucidate the process, this paper assumes that the input matrix obtained from the previous hidden layer is denoted by h, and the output Gaussian distribution parameters are denoted by θ. The expectation *μ* and standard deviation *σ* of the output can be expressed as follows:(11)$$\mu =\omega_{1}h+b_{1}$$
(12)$$\sigma =\mathrm{softplus}\left(\omega_{2}h+b_{2}\right)$$
where $\omega_{1}$ and $\omega_{2}$ represent the weight coefficient matrices of the output parameters of the probability density estimation layer, and $b_{1}$ and $b_{2}$ denote the bias matrices. To ensure a non-negative standard deviation, the softplus activation function is employed. The probability density estimation layer receives inputs including ambient temperature, VIC, and deflection at the previous time step. These inputs are integrated by a multilayer neural network, and the final output comprises the Gaussian distribution parameters of the deflection at the current time step. Consequently, the model outputs not only prediction intervals with different confidence levels but also deterministic prediction curves based on the expectation of the Gaussian distribution.

### 2.4. Evaluation Metrics

The model presented in this paper provides both interval and deterministic prediction results. To assess the accuracy of the deterministic predictions, this paper employs the Mean Squared Error (MSE), Root Mean Squared Error (RMSE), and the coefficient of determination ($R^{2}$) assessment indicators. These metrics are calculated as follows [33]:(13)$$\mathrm{MSE}=\frac{1}{m}{\sum_{i=1}^{m}\left(y_{i}-{\hat{y}}_{i}\right)}^{2}$$
(14)$$\mathrm{RMSE}=\sqrt{\frac{1}{m}{\sum_{i=1}^{m}\left(y_{i}-{\hat{y}}_{i}\right)}^{2}}$$
(15)$$R^{2}=1-\left\{\frac{{\sum_{i=1}^{m}\left(y_{i}-{\hat{y}}_{i}\right)}^{2}}{{\sum_{i=1}^{m}\left(y_{i}-{\bar{y}}_{i}\right)}^{2}}\right\}$$
where the length of the training and test data for deflection is assumed to be *m*. $y_{i}$ represents the true deflection value obtained from monitoring. $\hat{y_{i}}$ denotes the predicted value of deflection obtained by training the network, and $\bar{y_{i}}$ signifies the mean value of deflection.

This paper utilizes Prediction Interval Coverage Probability (PICP) and Prediction Interval Normalized Average Width (PINAW) as metrics for assessing prediction intervals. Interval coverage refers to the probability that the actual deflection monitoring value falls between the upper and lower bounds of the deflection estimation interval predicted within a specific confidence interval. A higher interval coverage suggests that the model can predict the unknown data more accurately and with greater precision. The calculation formula for these metrics is as follows:(16)$$\mathrm{PICP}=\frac{1}{m}\sum_{i=1}^{i}Z\left(y_{i}\in \left[U_{i}^{\alpha },L_{i}^{\alpha }\right]\right)$$
(17)$$Z=\left\{\begin{array}{c} 1,y_{i}\in \left[U_{i}^{\alpha },L_{i}^{\alpha }\right]\\ 0,y_{i}\notin \left[U_{i}^{\alpha },L_{i}^{\alpha }\right] \end{array}\right.$$
where the length of the training and test data for the deflection is assumed to be m. $y_{i}$ represents the true deflection value obtained from the monitoring. $U_{i}^{\alpha }\;\mathrm{and}\;L_{i}^{\alpha }$ are the upper and lower limits of the probability prediction intervals at the 1−α confidence level, respectively. PINAW is defined as the average width of all prediction intervals, reflecting the degree of uncertainty in the model prediction. When the PICP is high, a smaller PINAW indicates that the model’s predicted intervals are more precise, with lower uncertainty and better predictive performance. The calculation formula is as follows:(18)$$\mathrm{PINAW}=\frac{1}{mR}\sum_{i=1}^{i}\left(U_{i}^{\alpha }-L_{i}^{\alpha }\right)$$
where R represents the difference between the maximum and minimum values of the true deflection value, which is used to normalize the average width.

### 2.5. Modeling Strategy

The process of establishing a probabilistic prediction model for the vertical deflection of a bridge is outlined as follows:(1)Extract the ambient temperature, vehicle, and deflection monitoring data from the bridge’s structural health monitoring system. Given that the vehicle data are discrete, they cannot be directly inputted into the probabilistic model. Hence, it is essential to convert the raw WIM monitoring data into time-continuous VIC by employing linear superposition of the deflection influence lines.(2)Normalize the VIC, ambient temperature, and deflection monitoring data using the following formula:
(19)$${\hat{x}}_{i}=\frac{x_{i}-\mu }{\sigma }$$
where $x_{i}$ and ${\hat{x}}_{i}$ are the original data and the normalized transformed data, respectively. μ is the mean of the overall sample space, and σ is the standard deviation of the overall sample space.(3)Construct an interval prediction model comprising an input layer, a CNN layer, a pooling layer, an LSTM layer, a fully connected layer, and a probability density estimation layer. The normalized data are divided into a training set and a test set. The training set is used for network training, while the test set is used to evaluate the accuracy of the prediction model.(4)Utilize temperature data, vehicle influence coefficient data, and deflection monitoring data from the previous moment as inputs to generate a Gaussian distribution parameter matrix for the bridge deflection data output, thereby providing a prediction value for the deflection along with its corresponding confidence interval. The likelihood function is computed through maximum likelihood estimation and employed as the loss function for training the deep learning model. Continuously adjust the model’s hyperparameters based on its prediction results to incrementally enhance its accuracy and robustness.

The main flow of the bridge vertical deflection probability prediction model constructed according to the above steps is shown in Figure 3.

## 3. Case Study: Deflection Prediction of the Main Beam for a Suspension Bridge

### 3.1. Bridge Overview

The Nanxi Suspension Bridge (NSB) serves as a pivotal control project of the Yilu Expressway, situated in the southern region of Sichuan Province, China. It boasts a single-span steel box girder suspension design with a main span measuring 820 m. To provide a comprehensive assessment of the NSB’s operational performance, an SHM system was meticulously installed on the bridge. This SHM system facilitates real-time monitoring of environmental conditions, structural response, and moving loads. Figure 4 illustrates the sensors utilized in this study.

(1)At the Luzhou Tower bridge location, a WIM system was installed to capture the arrival time, speed, and weight of four-lane vehicles at the bridgehead.(2)A temperature sensor is positioned at the center of the span to monitor the ambient temperature. This sensor records the temperature every minute.(3)The bridge employs a Connecting Pipe Monitoring System (CPS) to monitor its vertical deflection. This system involves connecting a reference point to a measurement point via a liquid-filled pipe. Any alteration in the position of the reference point and the measurement point leads to a change in the disparity in the liquid level height, thereby enabling the sensors to detect the vertical deflection at the measurement point. The system comprises 15 pressure transmitters and a water tank. One pressure transmitter is positioned as a reference point (RP) alongside the water tank in the Yibin tower. A total of seven pairs are formed by the remaining 14 pressure transmitters, which are positioned upstream and downstream at locations that correspond to *l*/8, *l*/4, 3*l*/8, *l*/2, 5*l*/8, 3*l*/4, and 7*l*/8 of the main beams. The pressure transmitters operate at a sampling frequency of 0.5 Hz.

### 3.2. Data Preprocessing

The inputs to the neural network model comprised ambient temperature data, VIC, and deflection at the previous moment. The deflection monitoring data are collected once every 2 s, resulting in a total of 43,200 data points per day. Meanwhile, the data collection interval for ambient temperature is set to 1 min, requiring linear interpolation of the collected ambient temperature data, as depicted in Figure 5a. Recognizing that the vehicle load data obtained from the WIM system are discrete while the deflection monitoring data acquired by the CPS are time-continuous, this paper employs the approach proposed by Deng et al. [28] to convert discrete vehicle-load data into continuous VIC through the linear superposition of deflection influences. Figure 5b illustrates the VIC obtained from the middle of the span. Furthermore, the deflection monitoring data utilize the average of the monitoring values from upstream sensor P4-1 and downstream sensor P4-2 within the span. Figure 5 depicts the ambient temperature time series, which exhibits a typical daily variation pattern with a noticeable time lag effect relative to the deflection time series. Conversely, the VIC time series displays random fluctuations around zero, suggesting that the short-term fluctuations of the deflection data are attributable to the vehicle-load effect, while the long-term fluctuations are influenced by the ambient temperature.

To investigate the influence of the vehicle-load effect, ambient temperature, and other uncertainties on deflection, simple methods like linear regression are insufficient to capture the intricate correlations present. Therefore, this paper utilizes a CNN-LSTM-GD probabilistic model for interval prediction of deflection. A comparative study is carried out with the CNN-LSTM model and LSTM model.

### 3.3. Model Parameter Setting

To enhance the model’s accuracy, continuous adjustment of the network structure’s hyperparameters is imperative. The hyperparameters of the three models are presented in Table 1. The time step i of the time-series model is a crucial hyperparameter significantly affecting network performance. To ascertain the optimal time step, time steps ranging from i=1 to i=30 were substituted into the CNN-LSTM-GD, CNN-LSTM, and LSTM networks, and the outputs were evaluated using MSE. As depicted in Figure 6, the results indicate that i = 15 is the optimal input time step for the network. To prevent the model from converging to a local optimum, the gradient decay method is employed. The learning rate diminishes from 0.0001 to 0.00005 after 100 epochs of training the network structure.

### 3.4. Deflection Interval Prediction Model

Given that monitored deflections in the field stem from ambient temperature, vehicle loads, and assorted uncertainties, merely modeling deflections with high accuracy based on ambient temperature and vehicle loads is insufficient. Short-term variations in deflection sequences are linked to the vehicle-load effect, while long-term variations are correlated with ambient temperature. Consequently, it is evident that deflection monitoring data exhibit temporal correlation. Moreover, the deflection of the main girder may be influenced by the data point itself from the preceding moment. Thus, leveraging historical deflection data serves to supplement incomplete ambient temperature and vehicle-load information and compensate for various uncertainties. To achieve this, the ambient temperature sequence, VIC sequence, and deflection sequence from the previous moment are the input, with the subsequent moment’s deflection being the output. The mapping relationship is depicted in Figure 7, with 15 input steps.

To assess the predictive performance of the probabilistic model across various time scales, the data from 12:00 to 13:00 in Figure 6 were designated as T1, comprising 1800 data points, while T2 represents the entire day’s monitoring data, comprising 43,200 data points. For T1, the training set encompasses the initial 70% of the data, with the remaining 30% reserved for testing the trained probabilistic model’s accuracy. Likewise, for T2, the first 80% of the data is utilized for training, while the last 20% is allocated for testing. Figure 8 illustrates the model’s loss after 500 iterations, indicating convergence and the production of a more suitable network model. 

To visually compare and analyze the performance of the three models in predicting T1 short-term deflection data, Figure 9 presents the deterministic prediction results of the LSTM and CNN-LSTM models alongside the deterministic and probabilistic prediction results under the 95% confidence intervals of CNN-LSTM-GD. It is observed that all three models exhibit errors in the time window where deflection fluctuation is minimal. Notably, the LSTM model demonstrates the largest error in the prediction results, while the CNN-LSTM-GD displays the smallest error. Moreover, the CNN-LSTM-GD model excels in capturing deflection extremes, accurately quantifying uncertainty with monitored values falling within its 95% confidence interval. To evaluate prediction performance, RMSE and $R^{2}$ are utilized to measure model loss and correlation, while PICP and PINAW are employed to assess model probability intervals. Specific results are presented in Table 2: the CNN-LSTM-GD model achieves an RMSE of 1.14 mm and a coefficient of determination of 0.9557, surpassing the LSTM model, which yields an RMSE of 2.16 mm and a coefficient of determination of only 0.8523. This underscores the probabilistic model’s superior prediction accuracy and correlation compared to the other two models. However, the probabilistic model not only predicts deterministic outcomes but also excels in quantifying uncertainty intervals. With PICP and PINAW values of 0.9924 and 0.4404, respectively, the model demonstrates a strong capability in accurately capturing uncertainty while maintaining a narrow interval width.

As the time scale increases, the correlation between the data sequences decreases, resulting in a reduction in the network’s prediction accuracy. To further investigate the accuracy of the deterministic prediction results of the CNN-LSTM-GD model, we analyzed the prediction results of one day’s long-time deflection data, as depicted in Figure 10. Table 3 illustrates that the loss errors of all three models escalate with the increasing time span. However, the RMSE of the probabilistic model stands at 1.97, which is the smallest among the three models. Correspondingly, its coefficient of determination reaches 0.9683, the highest among the three models. These findings underscore the superior accuracy of the probabilistic network model, rendering it practical for bridge monitoring operations. Furthermore, the model’s 95% confidence interval effectively quantifies uncertainty, with PICP and PINAW values of 0.9847 and 0.1120, respectively. This demonstrates that the model achieves a high interval coverage rate while maintaining a minimal interval width.

Given the significance of monitoring deflection extremes over specific time intervals in the operational oversight of large-span bridges, our analysis focuses on extracting maximum and minimum deflection values within a 1 min time window from the T2 test set data. This approach ensures comprehensive coverage and captures pivotal deflection variations. This process yields a total of 288 deflection extreme value sequences, facilitating a thorough examination of critical deflection trends and patterns.

In Figure 11, the prediction outcomes for the extreme values are illustrated, with the monitored and forecasted extreme deflection values shown on the horizontal and vertical axes, respectively. The deflection extremes are meticulously analyzed and computed based on the scatter plot. Notably, the coefficient of determination, $R^{2}$, approaches 1 in the figure, suggesting a robust relationship between the monitored and forecasted deflection extremes. Further calculations reveal the RMSE of the predicted sequence of deflection extremes, resulting in RMSE values of 1.7047 and 1.5668 for the maximum and minimum deflection values, respectively. It is worth mentioning that, in comparison to directly calculating the entire T2 deflection prediction sequence, the error in the extreme deflection prediction sequence is significantly decreased. Therefore, the CNN-LSTM-GD probabilistic neural network maintains robust prediction capabilities for all-day deflection extreme data.

### 3.5. Model Application

(1)Identification of abnormal deflection

Under prolonged exposure to traffic loading and temperature variations, the structural components of suspension bridges, such as cables and suspension rods, may incur damage, consequently impacting the deflection of the main girder. Detecting anomalies at such minute deflection levels poses a challenge with conventional monitoring techniques. However, leveraging a high-precision probabilistic network model holds promise for identifying such deviations. To evaluate the model’s efficacy in structural condition assessment, we simulated structural deterioration by introducing a 20% deviation to the deflection values of the test set within the T2 dataset. This deviation commenced at 80% of the test set duration. Figure 12 illustrates the test set with this introduced bias.

The biased test set was inputted into the pre-trained CNN-LSTM-GD probabilistic network model, and the resulting deterministic outputs were examined by computing the ratio of the original values to the predicted values, denoted as *R*. The outcomes of the biased output are depicted in Figure 13. Notably, when there is no bias, the *R* value remains relatively stable around 1.0. However, upon the commencement of bias at 80%, an immediate alteration in the ratio occurs at the insertion point of the bias. This shift is attributed to the network’s mapping relationship, which aims to map data at the subsequent moment. The calculated value of *R* at 1.246 unmistakably indicates a bias of 20%. Subsequently, as the biased data replace the original data within the historical dataset, altering the temporal relevance of the input data, the time correlation properties gradually normalize, leading to the stabilization of R around 1.2.

Based on the provided information, while the model can initially identify abnormal deflections, it is essential to acknowledge that large-span bridges are subject to various uncertain factors, leading to significant fluctuations in model outputs during actual operation. Hence, solely relying on the ratio R for assessment may not offer sufficient reliability. To enhance real-time bridge monitoring and ensure safety and reliability, additional measures are imperative. Establishing a deflection warning threshold is one such measure. This threshold could be set based on historical data analysis and expert judgment. When the deflection values exhibit a discernible trend toward reaching a predefined threshold, timely warnings can be issued, prompting proactive intervention to mitigate potential risks.

(2)Early warning of structural state anomalies based on probabilistic interval prediction

As per the Technical Specification for Structural Safety Monitoring System of Highway Bridges (JT/T 1037-2022) [47], bridge safety warnings are categorized into two levels: yellow and red. A yellow warning serves to prompt the bridge management unit to closely monitor environmental conditions, loads, and structural responses, either holistically or locally. It necessitates enhanced tracking and observation efforts. Conversely, a red warning signifies a critical alert for the bridge maintenance unit. It prompts immediate attention to environmental factors, loads, and structural responses. The maintenance team must swiftly identify the cause of the alarm and undertake appropriate inspection and emergency management measures to ensure the bridge’s safe operation. Additionally, a timely structural safety assessment is imperative. According to the specification, a yellow warning is warranted when displacement or deformation exceeds 0.8 times the design value. A red warning is issued if the displacement or deformation surpasses the design value or if there are more than 10 yellow warnings within a month. These thresholds help guide proactive maintenance and management actions, ensuring the structural integrity and safety of the bridge under varying operating conditions.

The appropriateness of early warning thresholds directly impacts the effectiveness of a monitoring system, as an unreasonable threshold can lead to inefficient resource allocation. Thus, determining the optimal early warning threshold is crucial. Common approaches for setting these thresholds include statistical analysis, finite element analysis, historical data analysis, and standard specification analysis [48,49,50]. While statistical and finite element methods can yield complex equations, the probabilistic neural network offers a simpler and more convenient alternative, providing uncertainty intervals as deflection thresholds that can adjust with changing monitoring data. Following the “σ-principle” within the normal distribution, this study established two levels of warning mechanisms, as depicted in Figure 14:(1)The primary warning level employs the boundary of the (μ+σ,μ−σ) interval as the warning threshold. Given that the probability of a value falling within the (μ+σ,μ−σ) interval is 0.6826, an abnormal situation should be reported if 32% of the actual monitored deflection values fall outside this interval, prompting the maintenance unit to enhance supervision and inspection.(2)The secondary warning level utilizes the boundary of the (μ+2σ,μ−2σ) interval as the warning threshold. The probability of a value falling within the (μ+2σ,μ−2σ) interval is 0.9544. If 5% of the actual monitored deflection values exceed this interval, the relevant maintenance unit should closely monitor structural changes over an extended period, investigate the cause of the alert, ensure the operational safety of the bridge structure, and conduct a safety assessment of the bridge.

Therefore, the probability density interval estimation neural network can not only identify abnormal deflections through deterministic predictive values but also conduct deflection early warning assessments based on its interval distribution.

## 4. Conclusions

This paper presents a probabilistic model for predicting deflection intervals with high accuracy. The model is developed using a combination of a CNN, an LSTM neural network, and a probability density estimation layer. The main conclusions are as follows:(1)A probabilistic model for predicting deflection intervals is proposed, leveraging a CNN, LSTM, and Gaussian distribution probability density functions. The model takes into account temperature monitoring data, vehicle influence coefficients (VICs), and deflection monitoring data from the previous moment as inputs. Subsequently, it generates predictions for both the mean and standard deviation of deflection monitoring data at the next moment. This approach offers the advantage of providing not only deterministic predictions but also quantification of uncertainty, enabling the calculation of various confidence intervals. By leveraging this methodology, the model effectively addresses the complex nonlinear relationship between deflection, vehicle load, and temperature, making it well suited for comprehensive analysis in structural health monitoring applications.(2)This study evaluated the predictive performance of the probabilistic neural network model, revealing that the CNN-LSTM-GD model exhibited superior generalization ability and accuracy compared to the LSTM and CNN-LSTM models across various time scales. Specifically, for short-term deflection monitoring data characterized by minimal fluctuations, the CNN-LSTM-GD model accurately predicted the time-domain waveforms of the deflection data. Similarly, for long-term deflection monitoring data, the CNN-LSTM-GD model demonstrated enhanced predictive capability, particularly in forecasting extreme deflection within specific time windows. Additionally, the model’s 95% confidence intervals effectively quantify uncertainty, maintaining both a high interval coverage rate and a narrow interval width.(3)By introducing bias to the deflection input data and leveraging the deterministic prediction outcomes from the probabilistic model, anomalies in bridge operation can be more effectively identified through the ratio R. Furthermore, two warning mechanisms have been devised using the uncertainty intervals generated by the probabilistic model to establish deflection warning thresholds. These methods offer greater efficiency and convenience compared to traditional statistical analysis and finite element methods, enabling timely assessment of the bridge’s safety condition.

The main limitation of this work is that the proposed model only considers deflection prediction in a single section. In most cases, it is necessary to predict deflections in multiple sections rather than just one. It is important to note that there are temporal and spatial correlations between the deflections in different sections of a bridge. Therefore, further exploration of models that account for these correlations is required to enable multi-section deflection prediction. Additionally, in our application scenario, there is a large amount of measured data available. If there is an insufficient sample size in certain scenarios, it becomes challenging to ensure the generalization ability of the model. Therefore, further investigation is needed into deep learning strategies for small sample datasets, such as data augmentation and transfer learning.

## Figures and Tables

**Figure 1 sensors-24-06863-f001:**
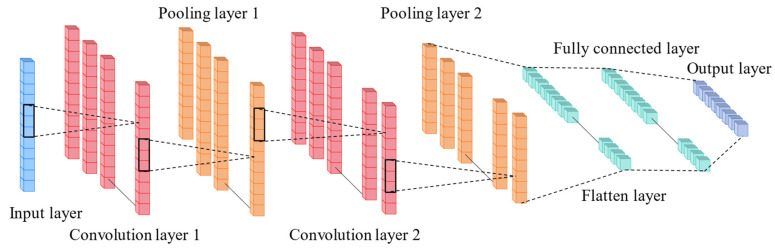
One-dimensional convolution neural network structure.

**Figure 2 sensors-24-06863-f002:**
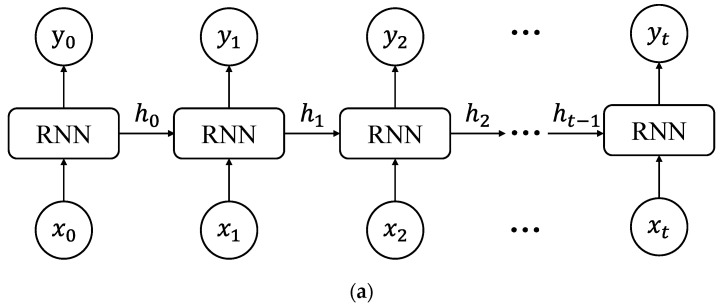
Neural networks and unit structures. (**a**) Recurrent Neural Network. (**b**) Long Short-Term Memory neural network. (**c**) Long Short-Term Memory unit.

**Figure 3 sensors-24-06863-f003:**
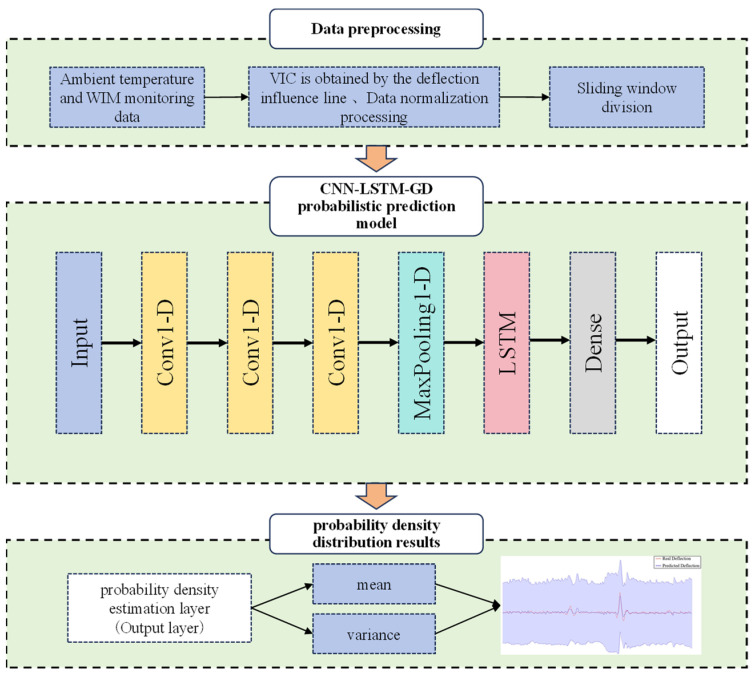
Modeling strategy of bridge deflection.

**Figure 4 sensors-24-06863-f004:**
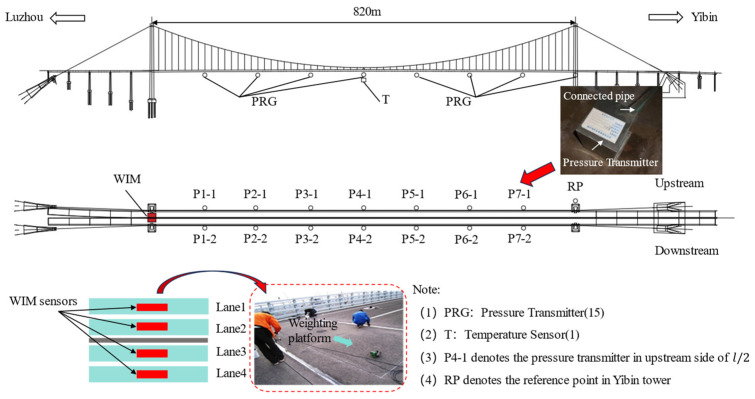
WIM system and deflection sensors of NSB.

**Figure 5 sensors-24-06863-f005:**
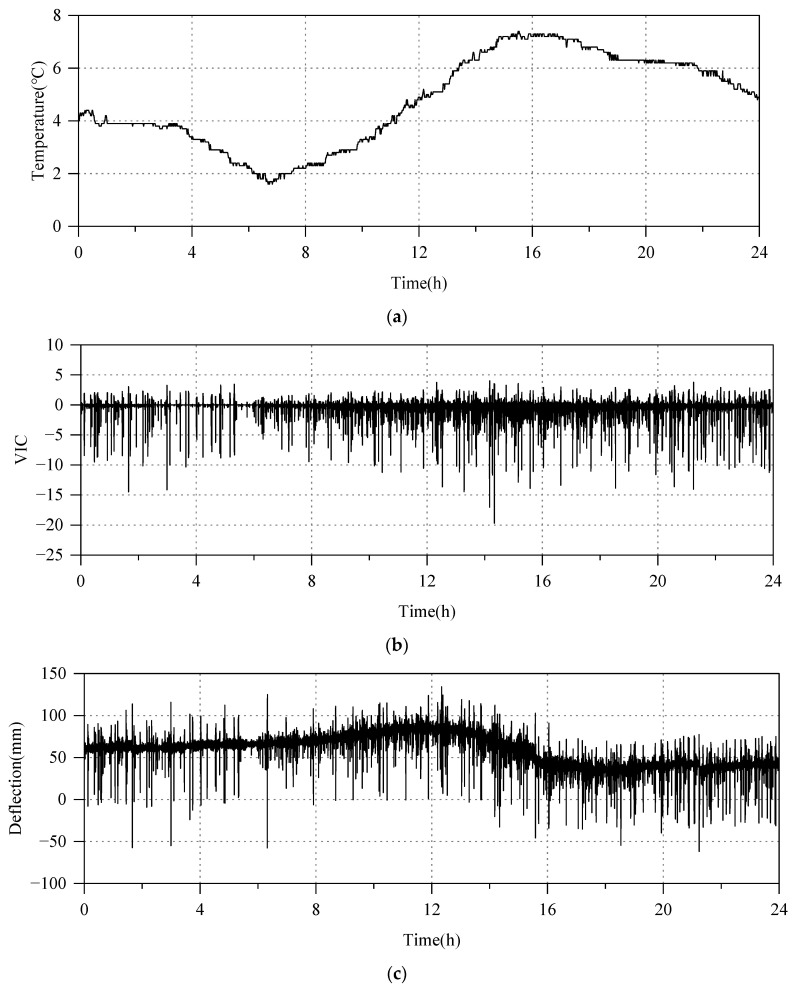
The monitored data from the NSB’s monitoring system. (**a**) Environmental temperature. (**b**) VIC of section l/2. (**c**) Deflection of section l/2.

**Figure 6 sensors-24-06863-f006:**
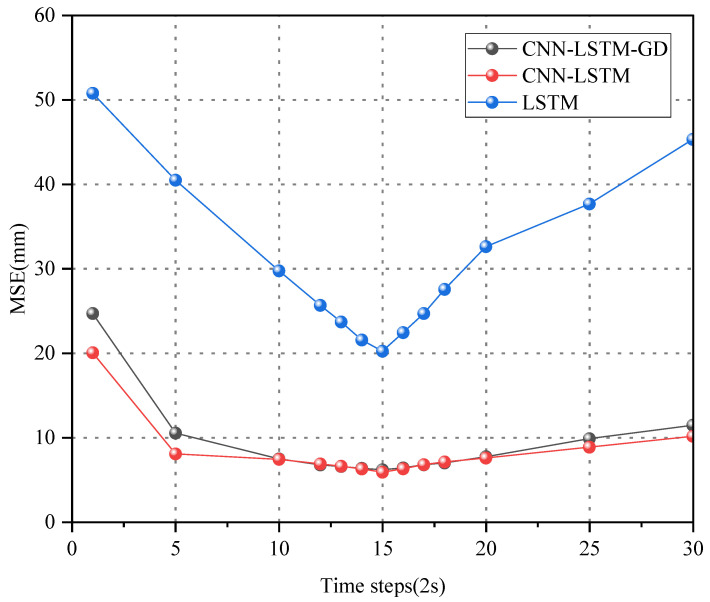
Relation between input steps and output error.

**Figure 7 sensors-24-06863-f007:**
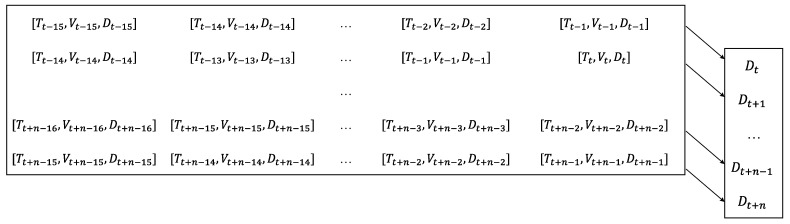
Mapping relationships between network model inputs and outputs.

**Figure 8 sensors-24-06863-f008:**
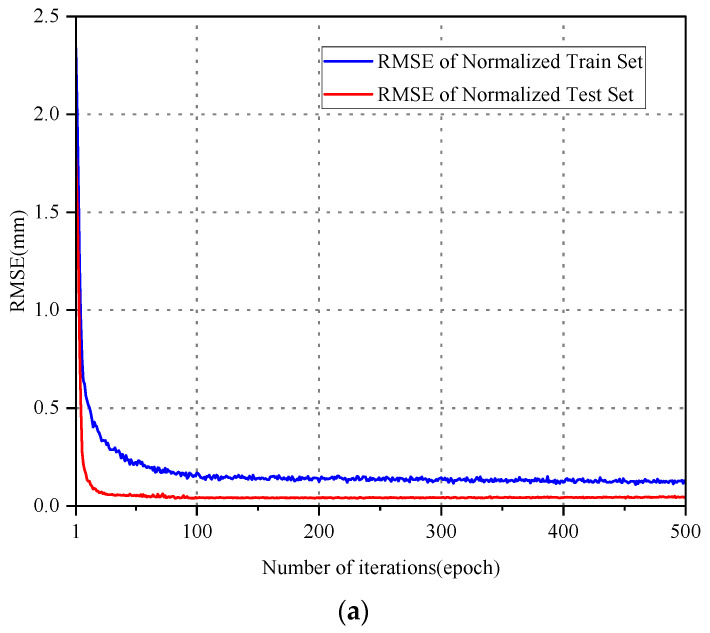
Loss curves of neural networks. (**a**) LSTM. (**b**) CNN-LSTM. (**c**) CNN-LSTM-GD.

**Figure 9 sensors-24-06863-f009:**
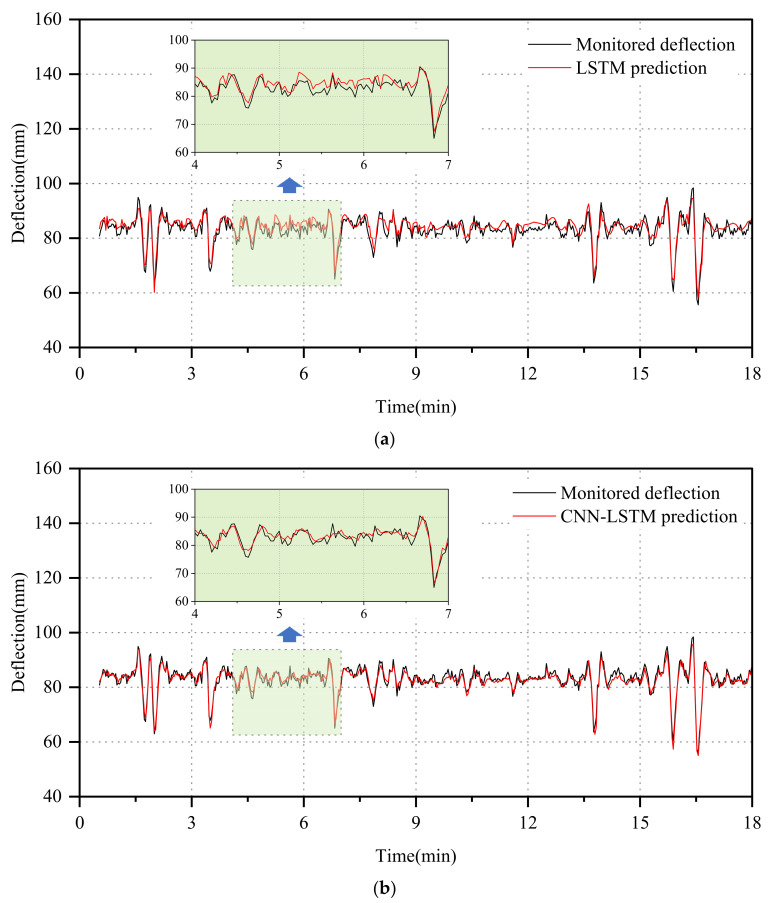
Prediction results of T1 monitoring data. (**a**) LSTM. (**b**) CNN-LSTM. (**c**) CNN-LSTM-GD.

**Figure 10 sensors-24-06863-f010:**
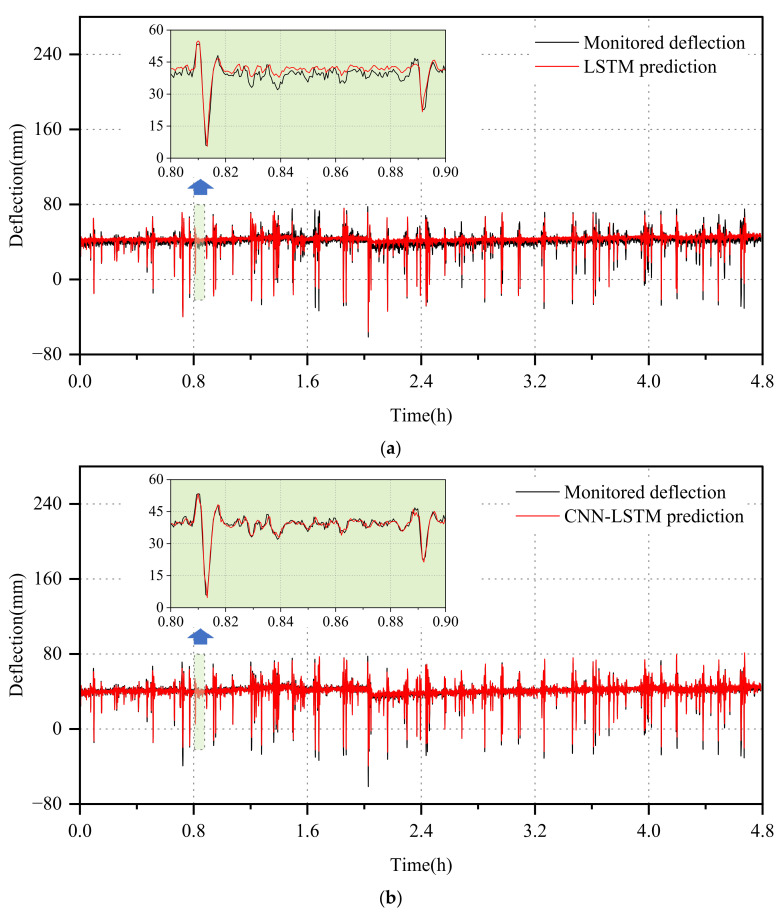
Prediction results of T2 monitoring data. (**a**) LSTM. (**b**) CNN-LSTM. (**c**) CNN-LSTM-GD.

**Figure 11 sensors-24-06863-f011:**
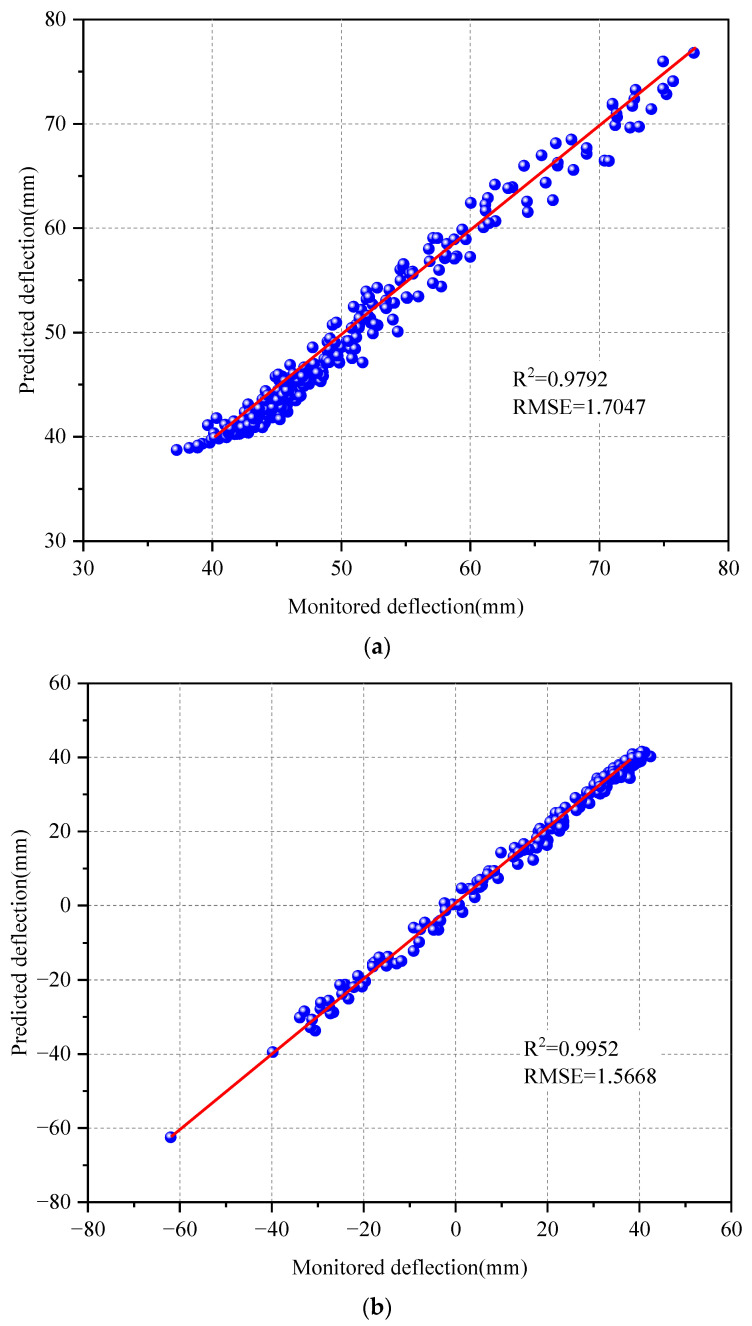
Prediction results of extreme values of T2 monitoring data. (**a**) Maximum. (**b**) Minimum.

**Figure 12 sensors-24-06863-f012:**
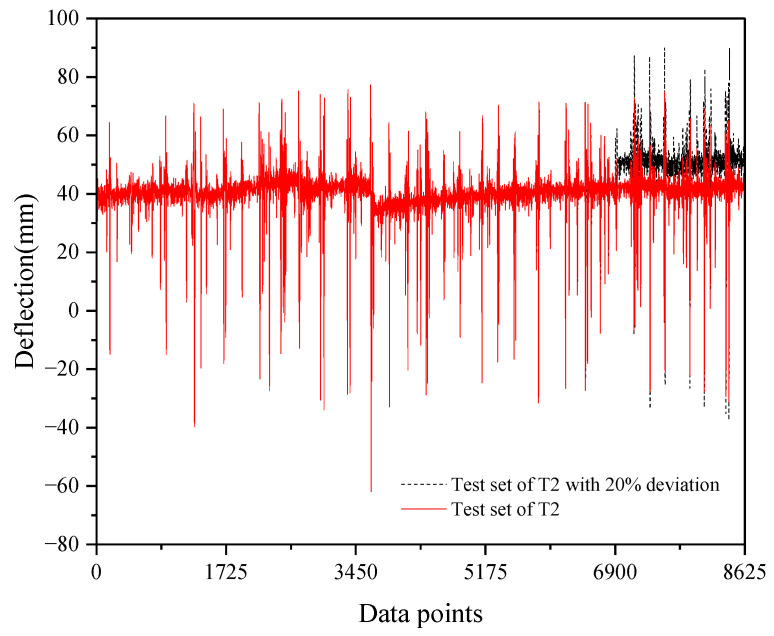
Test set with deviation from 80%.

**Figure 13 sensors-24-06863-f013:**
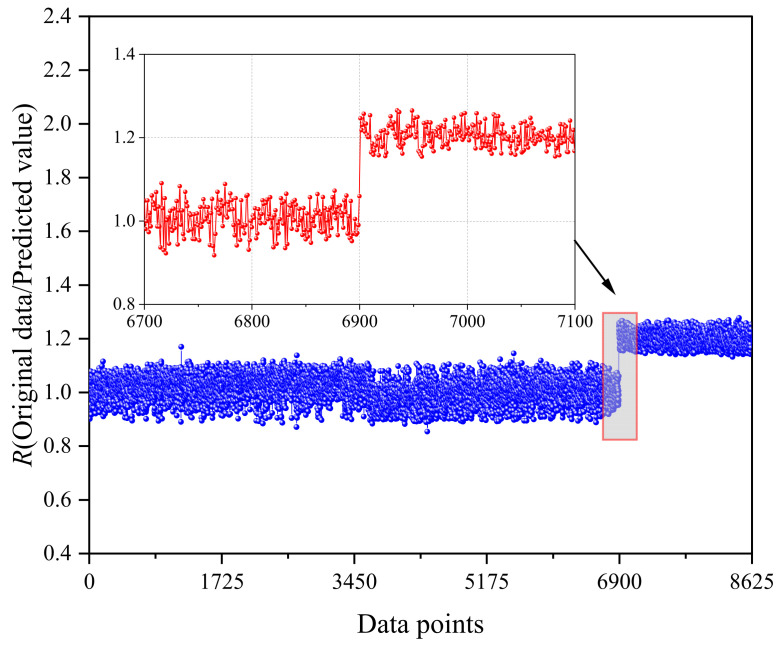
Output of test set with deviation from the 80% position.

**Figure 14 sensors-24-06863-f014:**
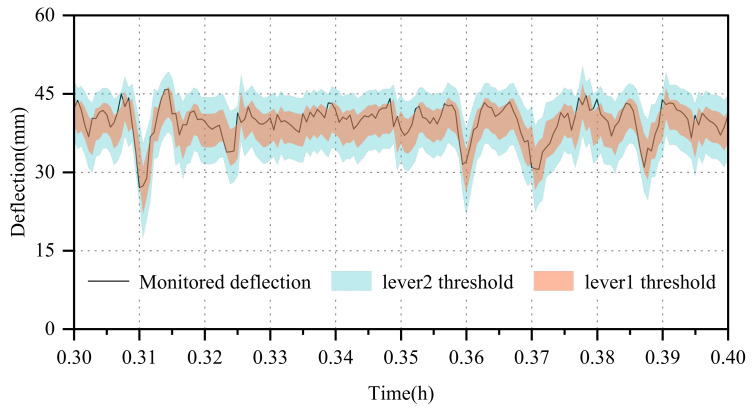
Mid-deflection threshold line.

**Table 1 sensors-24-06863-t001:** Hyperparameters of LSTM, CNN-LSTM, and CNN-LSTM-GD networks.

Category of Neural Network	The Number of Neurons in Convolutional Layer	The Number of Neurons in LSTM Layer	The Number of Neurons in Fully Connected Layer	Batch Size	Epoch	Learning Rate (lr)	Time Steps
CNN-LSTM-GD	128	128	2	512	500	0.0001–0.00005	15
CNN-LSTM	128	128	1	512	500	0.0001–0.00005	15
LSTM	/	128	1	512	500	0.0001–0.00005	15

**Table 2 sensors-24-06863-t002:** Prediction errors of T1 monitoring data.

Data	Kind	RMSE	$R^{2}$	PICP	PINAW
T1	LSTM	2.16	0.8523	/	/
	CNN-LSTM	1.59	0.8988	/	/
	CNN-LSTM-GD	1.14	0.9577	0.9924	0.4404

**Table 3 sensors-24-06863-t003:** Prediction errors of T2 monitoring data.

Data	Kind	RMSE	$R^{2}$	PICP	PINAW
T2	LSTM	4.32	0.8785	/	/
	CNN-LSTM	3.20	0.9195	/	/
	CNN-LSTM-GD	1.97	0.9683	0.9847	0.1120

## Data Availability

Some or all data, models, or code that support the findings of this study are available from the corresponding author upon reasonable request.

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
