# Peer review of "A Novel Method of Bridge Deflection Prediction Using Probabilistic Deep Learning and Measured Data"

_sensors, 2024, doi:10.3390/s24216863_

Round 1

Reviewer 1 Report

Comments and Suggestions for Authors
  1. There are a few grammatical errors and non-standard expressions in the paper, such as the mismatch between ‘firstly’ on line 37 and ‘second’ on line 48. The authors are requested to carefully review the entire text and make necessary corrections.
  2. The CNN structure shown in Figure 1 was not adopted in the paper. If there is no compelling reason, it can be removed. Figure 2 should be moved to the end of the main text.
  3. Abbreviations should be accompanied by their full forms when first introduced in the paper, and can be used as abbreviations thereafter. For example, Vehicle Influence Coefficients (VIC). The authors are requested to carefully review the entire text.
  4. The introduction of CNN and CNN-LSTM is overly extensive. The authors are advised to simplify it. Additionally, unnecessary formulas should not be included in the paper.
  5. As the authors mentioned, CNN has been widely applied in health monitoring of civil engineering. However, the discussion on this topic in the Introduction section is somewhat insufficient. Several aspects, including response reconstruction (Reconstruction of structural acceleration response based on CNN-BiGRU with squeeze-and-excitation under environmental temperature effects), missing measurement data recovery methods (Missing measurement data recovery methods in structural health monitoring: The state, challenges, and case study), damage identification and model updating (Damage identification of steel bridge based on data augmentation and adaptive optimization neural network), and response prediction (Nonlinear modeling of temperature-induced bearing displacement of long-span single-pier rigid frame bridge based on DCNN-LSTM), need to be appropriately supplemented.
  6. On line 317, the lowercase letter ‘l’ can be italicized to distinguish it from the Arabic numeral ‘1’.
  7. On line 325, the expression ‘at a frequency of 1 minute’ is inaccurate. The author intended to indicate the sampling time interval, which should be corrected.
  8. In Figure 6, the author states that there is a lag effect between temperature and deflection. The reviewer believes that temperature is the cause and deflection is the effect, so it is inappropriate to say that there is a lag effect between temperature and deflection. The author is requested to provide an explanation.
  9. Additionally, why does temperature affect the deflection of suspension bridges, and what is the mechanism behind this? This deeper reason is more important, and the author is requested to provide an explanation.
  10. In Table 1, the hyperparameter ‘The number of neurons in fully connected layer’ is 2 in CNN-LSTM-GD but 1 in the other two models. The author is requested to explain this. Furthermore, the structure of Table 1 needs to be adjusted and can be arranged horizontally.
  11. On line 505, JT/T 1037-2016 has been superseded by the current standard JT/T 1037-2022. The author is requested to make the necessary correction.
Comments on the Quality of English Language
  1. There are a few grammatical errors and non-standard expressions in the paper, such as the mismatch between ‘firstly’ on line 37 and ‘second’ on line 48. The authors are requested to carefully review the entire text and make necessary corrections.

Author Response

We sincerely appreciate the revision opportunity provided by the editor and reviewers. The reviewers have provided valuable comments, which guide us to strengthen the presentation of the paper. Now, the manuscript has been carefully revised following the reviewer’s suggestion. Details of the response are provided below.

On behalf of all co-authors

Haiping Zhang

Hunan University of Technology

Comments#1: There are a few grammatical errors and non-standard expressions in the paper, such as the mismatch between ‘firstly’ on line 37 and ‘second’ on line 48. The authors are requested to carefully review the entire text and make necessary corrections.

Response #1: Thank you for your valuable feedback. The author sincerely apologizes for the poor quality of the English writing. The entire manuscript has been thoroughly revised to address the grammatical and structural issues. In addition, several professors in the relevant field have reviewed the revised version. We hope that this revision meets the writing standards.

Comments#2: The CNN structure shown in Figure 1 was not adopted in the paper. If there is no compelling reason, it can be removed. Figure 2 should be moved to the end of the main text.

Response #2: Thank you for your valuable feedback. The original manuscript indeed did not include the CNN structure shown in Figure 1, and the author has removed it in the revised version. The revisions have been highlighted in the manuscript for your review and confirmation.

Comments#3: Abbreviations should be accompanied by their full forms when first introduced in the paper, and can be used as abbreviations thereafter. For example, Vehicle Influence Coefficients (VIC). The authors are requested to carefully review the entire text.

Response #3: Thank you for your valuable feedback. The author has adjusted the use of abbreviations in the revised manuscript, providing the full form upon their first appearance and subsequently using the abbreviated form. This adjustment aims to ensure clearer and more accurate communication, enhancing the overall academic rigor of the article.

Comments#4: The introduction of CNN and CNN-LSTM is overly extensive. The authors are advised to simplify it. Additionally, unnecessary formulas should not be included in the paper.

Response #4: Thank you for your valuable feedback. As you pointed out, the introduction of CNN and CNN-LSTM was indeed too broad. We have revised this section to ensure that the article is more concise and focused. (Please refer to Lines 120-188 for details.)

Comments#5: As the authors mentioned, CNN has been widely applied in health monitoring of civil engineering. However, the discussion on this topic in the Introduction section is somewhat insufficient. Several aspects, including response reconstruction (Reconstruction of structural acceleration response based on CNN-BiGRU with squeeze-and-excitation under environmental temperature effects), missing measurement data recovery methods (Missing measurement data recovery methods in structural health monitoring: The state, challenges, and case study), damage identification and model updating (Damage identification of steel bridge based on data augmentation and adaptive optimization neural network), and response prediction (Nonlinear modeling of temperature-induced bearing displacement of long-span single-pier rigid frame bridge based on DCNN-LSTM), need to be appropriately supplemented.

Response #5: Thank you for your valuable feedback. This paper employs the CNN-LSTM model to predict bridge deflection. As you mentioned, the introduction did not provide a detailed discussion on the application of CNN in the field of civil engineering health monitoring. We have revised the introduction accordingly in the revised manuscript (see Lines 64-65 for details).

Comments#6: On line 317, the lowercase letter ‘l’ can be italicized to distinguish it from the Arabic numeral ‘1’.

Response #6: Thank you for your valuable suggestion. In accordance with your advice, we have changed the 'l' to italics at Line 309 of the revised manuscript.

Comments#7: On line 325, the expression ‘at a frequency of 1 minute’ is inaccurate. The author intended to indicate the sampling time interval, which should be corrected.

Response #7: Thank you for your valuable suggestion. As you pointed out, using the phrase "at a frequency of 1 minute" to describe the time interval is incorrect. We have made the necessary correction in the revised manuscript at Line 316.

Comments#8: In Figure 6, the author states that there is a lag effect between temperature and deflection. The reviewer believes that temperature is the cause and deflection is the effect, so it is inappropriate to say that there is a lag effect between temperature and deflection. The author is requested to provide an explanation.

Response #8: Thank you for your valuable suggestions. In response to the expert's feedback, the authors would like to provide the following explanation:

The literature (Deep Learning-Based Minute-Scale Digital Prediction Model of Temperature-Induced Deflection of a Cable-Stayed Bridge: Case Study) and (Correlation model of deflection, vehicle load, and temperature for in-service bridge using deep learning and structural health monitoring) have mentioned the potential existence of a time-lag effect between environmental temperature and temperature-induced deflection. Influenced by the deck, the upper part of the main girder typically warms up more quickly and induces structural responses earlier than the lower part. Additionally, wavelet decomposition of deflection monitoring data reveals a possible lag effect between temperature-induced deflection and environmental temperature, as illustrated in the following figure:

(1)temperature-induced deflection

(2)environmental temperature

Comments#9: Additionally, why does temperature affect the deflection of suspension bridges, and what is the mechanism behind this? This deeper reason is more important, and the author is requested to provide an explanation.

Response #9: Thank you for your valuable suggestions. In response to the expert's feedback, the authors would like to provide the following explanation:

The environmental temperature affects the deflection of cable-stayed bridges through several mechanisms:

  1. Thermal Expansion and Contraction: The main components of a cable-stayed bridge (such as steel cables, towers, and the deck) undergo thermal expansion or contraction with temperature changes. As the environmental temperature increases, the steel cables and bridge structure expand, leading to increased tension; conversely, a decrease in temperature causes contraction and reduced tension. This change in tension directly affects the overall load-bearing state of the bridge, resulting in variations in deflection.
  2. Changes in Prestress: Cable-stayed bridges typically rely on prestress to maintain structural stability. Temperature variations can cause prestress to relax or recover, thereby affecting the deflection of the bridge. For example, an increase in temperature may lead to the relaxation of prestressed cables, increasing the sag of the deck, while a decrease in temperature may restore some prestress, reducing deflection.
  3. Material Elastic Modulus: The elastic modulus of materials changes with temperature, impacting their stiffness. At high temperatures, the stiffness of materials decreases, making the bridge more susceptible to deflection; at low temperatures, the stiffness increases, resulting in relatively reduced deflection.
  4. Temperature Gradient Effect: Different parts of the bridge may experience temperature gradients, leading to uneven distribution of thermal stress. For instance, areas exposed to direct sunlight and shaded areas may have significant temperature differences, causing localized expansion or contraction and generating additional deflection.
  5. Rate of Environmental Temperature Change: Rapid temperature changes can induce thermal stress within materials, affecting the dynamic response and deflection of the bridge. In regions with large day-night temperature fluctuations, the bridge experiences significant deflection changes during these transitions.

In summary, environmental temperature influences the deflection of cable-stayed bridges through various mechanisms, including thermal expansion of components, changes in prestress state, variations in material stiffness, and the generation of thermal stress.

Comments#10: In Table 1, the hyperparameter ‘The number of neurons in fully connected layer’ is 2 in CNN-LSTM-GD but 1 in the other two models. The author is requested to explain this. Furthermore, the structure of Table 1 needs to be adjusted and can be arranged horizontally.

Response #10: Thank you for your valuable feedback. As you pointed out, the hyperparameter "The number of neurons in the fully connected layer" is set to 2 in the CNN-LSTM-GD model, while it is set to 1 in the other two models. This difference arises because the LSTM and CNN-LSTM models can only output the predicted deflection values and do not predict confidence intervals, whereas the CNN-LSTM-GD model outputs both the mean and variance of the predicted values. Therefore, the hyperparameter "The number of neurons in the fully connected layer" varies accordingly. Additionally, we have also adjusted the structure of Table 1 in the revised manuscript to ensure it is more aesthetically pleasing and concise (see Line 354).

Comments#11: On line 505, JT/T 1037-2016 has been superseded by the current standard JT/T

1037-2022. The author is requested to make the necessary correction.Thank you for your valuable feedback. We sincerely apologize for the incorrect use of the standard. Following your suggestion, we have changed JT/T 1037-2016 to JT/T 1037-2022 at Line 496.

Reviewer 2 Report

Comments and Suggestions for Authors

This paper introduces a deflection interval prediction model leveraging convolutional neural networks (CNN), long short-term memory (LSTM) networks, and a probability density estimation layer. Using a suspension bridge in China as a case study, the prediction accuracy was assessed across different time scales. The findings demonstrate that the probabilistic model exhibits superior prediction capabilities for both small deflection fluctuations and deflection extremes compared to the LSTM and CNN-LSTM models.

1.  In the abstract, we advised to rewrite in order to reflect the significance  according to Purpose, Method, Results, and Conclusion.

2.  In Introduction, authors have described the work of the paper at the end, but lacked the main contribution.  Please further summarize and clearly demonstrate the main contributions of your paper.

3. In the Section 2 of Deflection prediction model based on probabilistic deep learning, authors provided the overview of the Modeling strategy of bridge deflection. This is not enough. The control ideas, strategies, and methods of bridge deflection should be provided in detail.

4. In 3.5. Model Application, line 475-478, why did authors select 20% deviation?

5. In Conclusion, could you tell me the limitations of the proposed method?  Please add this part to the manuscript.

6. In order to highlight the introduction, some latest references should be added to the paper.  For example, https://ojs.bonviewpress.com/index.php/AIA/article/view/441; https://doi.org/10.1177/14759217241254121; https://doi.org/10.47852/bonviewAIA3202798 and so on.

7. More statistical methods are recommended to analyze the experimental results.

Comments on the Quality of English Language

The paper is in need of revision in terms of eliminating grammatical errors, and improving clarity and readability.

Author Response

We sincerely appreciate the revision opportunity provided by the editor and reviewers. The reviewers have provided valuable comments, which guide us to strengthen the presentation of the paper. Now, the manuscript has been carefully revised following the reviewer’s suggestion. Details of the response are provided below.

On behalf of all co-authors

Haiping Zhang

Hunan University of Technology

Comment #1: In the abstract, we advised to rewrite in order to reflect the significance according

to Purpose, Method, Results, and Conclusion.

Response #1:

Thank you for your valuable feedback; your suggestions are crucial for enhancing the quality of the paper. In response to your recommendations, we have refined and restructured the abstract in the revised manuscript to summarize the objectives, methods, results, and conclusions in a more concise and clear manner (see Lines 11-28 of the revised manuscript).

Comment #2: In Introduction, authors have described the work of the paper at the end, but lacked the main contribution. Please further summarize and clearly demonstrate the main contributions of your paper.

Response #2:

Thank you for your valuable feedback. As you pointed out, the introduction indeed lacked a clear statement of the main contributions. We have revised the introduction in accordance with your recommendations to explicitly outline the key contributions of the paper (see Lines 102-119).

Comment #3: In the Section 2 of Deflection prediction model based on probabilistic deep learning, authors provided the overview of the Modeling strategy of bridge deflection. This is not enough. The control ideas, strategies, and methods of bridge deflection should be provided in detail.

Response #3: Thank you for your valuable feedback. In Section 2, we have rewritten the modeling strategy for bridge deflection as per your request, providing a detailed explanation of the strategies and methods used for predicting bridge deflection (see Section 2.5).

Comment #4: In 3.5. Model Application, line 475-478, why did authors select 20% deviation?

Response #4: Thank you for your valuable feedback. The selection of a 20% deviation in the model application is based on the need to consider accuracy requirements. If a 10% deviation is chosen, most deflection deviations would be only 4 mm, which may not accurately identify abnormal deflections. Conversely, selecting a 30% deviation would not allow the model's potential to be fully realized. Therefore, to thoroughly understand the model's capabilities, this study has opted for a 20% deflection deviation.

Comment #5: In Conclusion, could you tell me the limitations of the proposed method? Please add this part to the manuscript.

Response #5: Thank you for your valuable feedback. As you pointed out, the conclusion section of the paper indeed lacks a discussion of the limitations of this work. The main limitation of this study is that the proposed model only considers deflection prediction for a single section. In many cases, it is necessary to predict the deflection of multiple sections rather than just one. There exists both temporal and spatial correlation between the deflections of different sections of the bridge. Moreover, there is a wealth of measurement data available in our application scenario. If the sample size is insufficient in certain cases, ensuring the model's generalization capability becomes very challenging. Therefore, further research is needed on deep learning strategies for small sample datasets, such as data augmentation and transfer learning. We have included this discussion in the revised manuscript (see Lines 581-590).

Comment #6: In order to highlight the introduction, some latest references should be added to the paper. For example, https://doi.org/10.47852/bonviewAIA320244;

https://doi.org/10.1177/14759217241254121;

https://doi.org/10.47852/bonviewAIA3202798 and so on.

Response #6: Thank you for your valuable feedback. As you pointed out, we indeed need to include the latest literature to enhance the introduction section. However, only the last DOI you provided was valid, and we have cited it in the paper (Reference 25).

Comment #7: More statistical methods are recommended to analyze the experimental results.

Response #7: Thank you for your valuable feedback. We acknowledge the need to employ additional statistical methods to analyze the experimental results. However, given that the current paper has already utilized RMSE, R², PICP, and PINAW as evaluation metrics for the statistical analysis of the model's predictions, and considering the need for reasonable allocation of the paper's length, we were unable to include a more detailed discussion of additional statistical methods in this manuscript. Nevertheless, we will certainly address this in future research endeavors.

Reviewer 3 Report

Comments and Suggestions for Authors

Comments to the Author(s)

This manuscript presents

 A Novel method of bridge deflection prediction using proba- bilistic deep learning and measured data. This paper contains a good effort related to the monitoring of

 Bridge infrastructures. The topic is original. The methodology of the study is well explained. In general, the manuscript is organized well. The study offers reliable findings and is supported by sufficient proof.  It could be accepted for publication if the authors resolve the following issues. All the answers should be included in the manuscript.

1.      Page 1, line 27, Instead of writing “ Structural Health Monitoring System (SHMS) …”, please write it as “ Structural Health Monitoring(SHM) system  …”

2.      Page 3, line 108, Instead of writing “ Convolutional Neural Networks  …”, please write it as “ Convolutional Neural Networks (CNNs)…”

3.      Page 4, line 142, Instead of writing “ The Long Short-Term Memory network  …”, please write it as “ The Long Short-Term Memory (LSTM) network …”

4.      Page 6, add a reference to the equations (1, 2, 3, 4, 5, 6).

5.      Page 7, add a reference to the equations (12, 13, 14).

6.      Page 9, line 302, I prefer to write “ SHMS  …” as “ SHM system …”

7.      The caption of Fig. 5. Rewrite the caption to “ WIM system and deflection sensors of NSB.”

8.      In section 3.4, I noticed that you divided the data into training and testing data only. What is about the validating data? However, I see validating data in figures only.

9.      Page 15, line 407. Instead of writing Table II, write it as Table 2.

10.  Figure 9. (Loss curves of neural networks), The CNNN-LSTM model has the minimum model's loss compared with the other two models. However, the story is different in Table 2. The CNN-LSTM-GD model has minimal Prediction errors. Please, explain that.

Author Response

We sincerely appreciate the revision opportunity provided by the editor and reviewers. The reviewers have provided valuable comments, which guide us to strengthen the presentation of the paper. Now, the manuscript has been carefully revised following the reviewer’s suggestion. Details of the response are provided below.

On behalf of all co-authors

Haiping Zhang

Hunan University of Technology

Comment #1: Page 1, line 27, Instead of writing “ Structural Health Monitoring System (SHMS) …”, please write it as “ Structural Health Monitoring(SHM) system …”

Response #1: Thank you for your valuable feedback. We sincerely apologize for our writing error. Following your suggestion, we have corrected the expression at Line 33 of the revised manuscript from "Structural Health Monitoring System (SHMS)" to the accurate "Structural Health Monitoring (SHM) system."

Comment #2: Page 3, line 108, Instead of writing “ Convolutional Neural Networks …”, please write it as“ Convolutional Neural Networks (CNNs)…”

Response #2: Thank you for your valuable feedback. We sincerely apologize for our writing error. In accordance with your suggestion, we have corrected the expression at Line 122 of the revised manuscript from "Convolutional Neural Networks" to the accurate "Convolutional Neural Networks (CNNs)."

Comment #3: Page 4, line 142, Instead of writing “ The Long Short-Term Memory network …”, please writeit as “ The Long Short-Term Memory (LSTM) network …”

Response #3:Thank you for your valuable feedback. We sincerely apologize for our writing error. Following your suggestion, we have corrected the expression at Line 146 of the revised manuscript from "The Long Short-Term Memory network" to the accurate "The Long Short-Term Memory (LSTM) network."

Comment #4、#5:

Page 6, add a reference to the equations (1, 2, 3, 4, 5, 6).

Page 7, add a reference to the equations (12, 13, 14).

Response #4、#5: Thank you for your valuable feedback. As you noted, the equations section indeed lacked relevant citations. In response, we have rigorously followed your recommendations to cite the appropriate related articles (see Lines 138、Line174、Line 231).

Comment #6: Page 9, line 302, I prefer to write “ SHMS …” as “ SHM system …”

Response #6: Thank you for your valuable feedback. We sincerely apologize for our writing error. Following your suggestion, we have corrected the expression at Line 293 of the revised manuscript from "SHMS ..." to the accurate "SHM system."

Comment #7: The caption of Fig. 5. Rewrite the caption to “ WIM system and deflection sensors of NSB.”

Response #7: Thank you for your valuable feedback. We have strictly followed your suggestion and revised the title of Figure 5 to "WIM system and deflection sensors of NSB" (see Line 312).

Comment #8: In section 3.4, I noticed that you divided the data into training and testing data only. What is about the validating data? However, I see validating data in figures only.

Response #8: Thank you for your valuable feedback. In response to your comments, we would like to clarify the following:

In this model, we did not create a validation set; instead, we divided the dataset into a training set and a testing set. The model is generated using the training set, and the testing set is used to assess the model's accuracy and error, thereby validating its effectiveness. This approach evaluates the model's robustness based on the predictive performance of the testing set, which is consistent with deep learning model training strategies. The mention of "validating set" in Figure 9 was a typographical error, and it has been corrected.

Comment #9: Page 15, line 407. Instead of writing Table II, write it as Table 2.

Response #9: Thank you for your valuable feedback. We sincerely apologize for our writing error. Following your suggestion, we have corrected the expression at Line 398 of the revised manuscript from "Table II" to the accurate "Table 2."

Comment #10: Figure 9. (Loss curves of neural networks), The CNNN-LSTM model has the minimum model'sloss compared with the other two models. However, the story is different in Table 2. The CNN-LSTM-GD model has minimal Prediction errors. Please, explain that.

Response #10: Thank you for your valuable feedback. In response to your comments, we would like to clarify the following:

Due to a typographical error, the first figure should represent the loss of the LSTM model, while the third figure depicts the loss of the CNN-LSTM-GD model. Both the LSTM and CNN-LSTM models use RMSE as their loss function, whereas the CNN-LSTM-GD model employs a custom negative log-likelihood function. Additionally, this loss function is normalized, resulting in values that are smaller than those presented in Table 2. The related errors have been corrected in the manuscript. We sincerely apologize for our mistakes.

Reviewer 4 Report

Comments and Suggestions for Authors

This paper presents a probabilistic model for predicting deflection intervals with high accuracy. This deflection interval prediction model leveraging convolutional neural networks(CNN), long short-term memory (LSTM) networks, and a probability density estimation layer. (CNN), long short-term memory (LSTM) networks, and a probability density estimation layer. By comparing the LSTM model, the CNN-LSTM model, and the CNN-LSTM-GD model, the CNN-LSTM-GD model demonstrates higher precision. The topic of the study is interesting, but there are many issues in this article that need further revision:

(1) Line117    What is the mean of excitation layer?

(2) Line178    The output gate text is not clear.

(3) Line334    The time lag in the figure is not obvious.

Author Response

We sincerely appreciate the revision opportunity provided by the editor and reviewers. The reviewers have provided valuable comments, which guide us to strengthen the presentation of the paper. Now, the manuscript has been carefully revised following the reviewer’s suggestion. Details of the response are provided below.

On behalf of all co-authors

Haiping Zhang

Hunan University of Technology

Comment #1: Line117 What is the mean of excitation layer?

Response #1: Thank you for your valuable feedback. In response to your comments, we would like to clarify the following:

In a CNN, the Activation Layer refers to the layer that applies a nonlinear activation function, typically following a convolutional layer or a fully connected layer. The primary purpose of the activation layer is to introduce non-linearity, enabling the neural network to learn and represent more complex patterns and features. Without the activation layer, a neural network would only be able to represent linear relationships, rendering it incapable of solving complex tasks. The roles of the activation layer are as follows:

  1. Introducing Non-linearity: Both convolutional and fully connected layers fundamentally perform linear operations (linear combinations and weighted sums). By incorporating an activation function, the output of these linear combinations can be transformed into non-linear forms, allowing the network to handle more complex data distributions.
  2. Enhancing Model Expressiveness: Activation functions can enhance the expressive capability of the model, allowing the network to learn features at different levels, from simple edges and textures to high-level object recognition.
  3. Preventing Network Degeneration: In the absence of activation layers, stacking multiple convolutional or fully connected layers results in merely a combination of linear transformations, which ultimately equates to a single-layer linear transformation. This implies that no matter how deep the network is, it loses the advantages of deep architectures. Activation layers are crucial for maintaining the effectiveness of deep networks.

Commonly used activation functions include ReLU, Tanh, Sigmoid, and Softmax.

Comment #2: Line178 The output gate text is not clear.

Response #2: Thank you for your valuable feedback. These comments are crucial for improving the quality of the paper. In response to your suggestions, we have refined and restructured the section on the 'output gate' in the revised manuscript, summarizing the structure of the LSTM unit in a more concise and clear manner (see revised manuscript, lines 164-166).

Comment #3: The time lag in the figure is not obvious.

Response #3: Thank you for your valuable feedback. In response to your comments, we provide the following explanation:

By using wavelet decomposition to extract the thermally induced deflection and comparing it with the environmental temperature data, we observed a significant lag effect. Considering that the authors aim to predict the monitored deflection values rather than the thermally induced deflection, we did not include the thermally induced deflection in the figure (as shown below). Therefore, we can conclude that there is indeed a temporal lag effect between the environmental temperature and deflection.

(1)temperature-induced deflection

(2)ambient temperature

Reviewer 5 Report

Comments and Suggestions for Authors

This paper proposes a method for predicting bridge deflection data, which has significant engineering application value.

However, several critical issues are not clearly articulated:

1) Why choose a network architecture based on CNN and LSTM over traditional machine learning algorithms? What advantages does it offer?

2) If there is spatial dependency (CNN), where is it reflected(feature or element)? CNNs are mostly used in vision, to extract spatial features, and as seen in the text, there is only one feature that is deflection.

3) If there is temporal dependency (LSTM), where is it reflected(feature or element)? LSTM is mostly used for this kind of time series processing and may be based on LSTM network design only process to get better results.

4) The paper does not mention the relationship between the duration of the samples used and the prediction time.

5) Assuming that the prediction results follow a normal distribution is highly imprecise. Conducting a normality test on the results before proceeding with further work is necessary.

6) Labeling parameter estimation for prediction results as 'probabilistic deep learning' is considered a misuse of the concept.

For these reasons, I recommend rejecting this paper.

Author Response

We sincerely appreciate the revision opportunity provided by the editor and reviewers. The reviewers have provided valuable comments, which guide us to strengthen the presentation of the paper. Now, the manuscript has been carefully revised following the reviewer’s suggestion. Details of the response are provided below.

On behalf of all co-authors

Haiping Zhang

Hunan University of Technology

Comment #1: Why choose a network architecture based on CNN and LSTM over traditional machinelearning algorithms? What advantages does it offer?

Response #1: Thank you for your valuable feedback. In response to your comments, we provide the following explanation:

When it comes to processing time series data, choosing a network architecture based on Convolutional Neural Networks (CNN) and Long Short-Term Memory (LSTM) networks offers several important advantages over traditional machine learning algorithms:

  • Automatic Feature Extraction: The convolutional layers in CNNs can automatically learn and extract local features from the time series without the need for manual feature engineering. Traditional machine learning algorithms (such as SVM and random forests) often rely on hand-crafted features, which are time-consuming to develop and may not effectively capture the complexities of the data. LSTM networks excel at remembering long-term dependencies in time series data, making them particularly suitable for handling complex data with temporal characteristics.
  • Handling Nonlinear and Complex Relationships: Time series data often exhibits high levels of nonlinearity and complex dynamic relationships. The combination of CNN and LSTM networks, with their multi-layer nonlinear structures, enables better learning and modeling of complex nonlinear relationships. In contrast, traditional algorithms like linear regression and ARIMA models, while simple and intuitive, struggle to capture nonlinear relationships effectively.
  • Capturing Spatial and Temporal Information: CNNs are adept at extracting spatial patterns from local time windows (i.e., short-term trends or local temporal dependencies) through convolutional operations. LSTMs can manage long-term sequential dependencies thanks to their memory units and gating mechanisms, effectively retaining long-term information and addressing sequential dependency issues. Traditional machine learning methods typically operate based on fixed windows or statistical features, lacking the ability to capture multi-level spatial-temporal information as effectively as CNNs and LSTMs. For instance, random forests and decision trees are generally unable to effectively remember and utilize information from long time dependencies.
  • Performance Advantages: Research has shown that deep learning models based on CNNs and LSTMs tend to significantly outperform traditional machine learning methods in terms of accuracy and robustness, particularly when dealing with large datasets that exhibit complex temporal dependencies.

In summary, the advantages of using CNN and LSTM-based architectures for time series processing primarily lie in their ability for automatic feature extraction, modeling long and short-term dependencies, handling complex nonlinear relationships, and offering strong scalability and robustness. In contrast, traditional machine learning algorithms are more reliant on feature engineering and exhibit weaker performance in addressing complex time dependencies, making deep learning architectures often more suitable for large and complex time series data processing tasks.

Comment #2: If there is spatial dependency (CNN), where is it reflected(feature or element)? CNNs are mostly used in vision, to extract spatial features, and as seen in the text, there is only one feature that is deflection.

Response #2: Thank you for your valuable feedback. In response to your comments, we would like to clarify the following:

Although CNNs are primarily used for visual tasks, one-dimensional convolution also offers significant advantages in time series prediction. The convolution operations in CNNs can effectively handle multi-dimensional inputs, such as incorporating various vehicle and environmental temperature variables, rather than being restricted to different geographical or spatial locations of the same feature. CNNs excel at extracting spatial patterns from local time windows (i.e., short-term trends or local temporal dependencies) by learning local features through convolutional operations.

Comment #3: If there is temporal dependency (LSTM), where is it reflected(feature or element)? LSTM is mostly used for this kind of time series processing and may be based on LSTM network

design only process to get better results.

Response #3: Thank you for your valuable feedback. In response to your comments, we would like to clarify the following:

Temporal correlation refers to the interdependence of data along the time dimension. In other words, it assesses whether the data value at a particular time point is related to the values at previous or subsequent time points. In this paper, the sequences of vehicle load, environmental temperature, and deflection are all time series that may exhibit temporal correlation. Regarding your suggestion that LSTM could yield better results, we have conducted a comparison in the paper, and the results of the CNN-LSTM model outperform those of the LSTM model.

Comment #4: The paper does not mention the relationship between the duration of the samples used and the prediction time.

Response #4:Thank you for your valuable feedback. In response to your comments, we would like to clarify the following:

When you mentioned "the relationship between the duration of the samples used and the prediction time," are you referring to the training time and testing time? There isn't a specific correspondence between the two; typically, the testing time is much shorter than the

Comment #5: Assuming that the prediction results follow a normal distribution is highly imprecise.

Conducting a normality test on the results before proceeding with further work is necessary.

Response #5: Thank you for your valuable feedback. In response to your comments, we would like to clarify the following:

In this paper, we only assume that the deflection at each moment follows a normal distribution, rather than assuming that all data follows a normal distribution. This assumption is made to predict the uncertainty associated with the deflection data, indicating the likelihood that the predicted value falls within a certain range at a given moment, rather than suggesting that the deflection predictions themselves adhere to a normal distribution. Therefore, we believe it is reasonable to make this assumption.

Comment #6: Labeling parameter estimation for prediction results as 'probabilistic deep learning' is considered a misuse of the concept.

Response #6: The probabilistic deep learning methods mentioned in this paper are discussed in the book Probabilistic Deep Learning with Python, Keras, and TensorFlow Probability, authored by Oliver Duerr and Beate Sick, with ISBN 9781617296079 (available on Manning eBooks and IEEE Xplore). Therefore, the authors believe that it is appropriate to adopt the concept of probabilistic deep learning in this context.

Round 2

Reviewer 1 Report

Comments and Suggestions for Authors

Thank you for addressing my comments.

Author Response

Thanks.

Reviewer 2 Report

Comments and Suggestions for Authors

According to the revised paper, I have appreciated the deep revision of the contents and the present form of this manuscript. But there is still a little content, which need be revised according to the comment of reviewer in order to meet the requirements of publish. A number of concerns listed as follows:

1.  In the abstract section, I would suggest that the author should provide to the point and quantitative advantages of the proposed method.

2. The authors need to interpret the meanings of the variables, such as Eq.(8), (13),....

3. Please add the contents of Author Contributions, Institutional Review Board Statement, Informed Consent Statement, Data Availability Statement and Conflicts of Interest.

4. The invalid DOI is https://doi.org/10.47852/bonviewAIA3202441, https://doi.org/10.1016/j.engappai.2024.109237, https://doi.org/10.1109/JIOT.2024.3409823

5. There are a few typos and grammar errors in the manuscript. Please polish the manuscript carefully.

Comments on the Quality of English Language

There are a few typos and grammar errors in the manuscript. Please polish the manuscript carefully.

Author Response

We sincerely appreciate the revision opportunity provided by the editor and reviewers. The reviewers have provided valuable comments, which guide us to strengthen the presentation of the paper. Now, the manuscript has been carefully revised following the reviewer’s suggestion. Details of the response are provided below.

On behalf of all co-authors

Haiping Zhang

Hunan University of Technology

Comment #1: In the abstract section, I would suggest that the author should provide to the point and quantitative advantages of the proposed method.

Response #1: Thank you for your valuable feedback, these comments are crucial for improving the quality of the paper. In response to your suggestions, we have refined and restructured the abstract in the revised manuscript to quantify the superiority of the proposed model in a more concise and clear manner. (See revised manuscript Lines 23-28 for details.)

Comment #2: The authors need to interpret the meanings of the variables, such as Eq.(8), (13),....

Response #2: Thank you for your valuable feedback. In response to your comments, we provide the following explanations:

  1. Equation (8) represents the objective of minimizing the negative log-likelihood function, where the likelihood function is given in Equation (9), serving as the probability density function for deflection. The variables are explained further in Lines 208-222 of the manuscript.
  2. In Equation (13), the Mean Squared Error (MSE) refers to the average of the squared differences between the predicted values and the actual observations.

We appreciate your suggestions once again. We have reviewed the entire manuscript and clarified the explanations for the relevant formula variables.

Comment #3: Please add the contents of Author Contributions, Institutional Review Board Statement, Informed Consent Statement, Data Availability Statement and Conflicts of Interest.

Response #3: Thank you for your valuable feedback. We have added the Author Contributions, Data Availability Statement, Informed Consent Statement and Conflicts of Interest at the end of the revised manuscript. However, this thesis does not include an Institutional Review Board Statement.

Comment #4: The invalid DOI is https://doi.org/10.47852/bonviewAIA3202441,https://doi.org/10.1016/j.engappai.2024.109237,https://doi.org/10.1109/JIOT.2024.3409823

Response #4:Thank you for your valuable feedback. We acknowledge the need to cite relevant recent literature, and we have included citations to these sources in the revised manuscript (References 38-40).

Comment #5: There are a few typos and grammar errors in the manuscript. Please polish the manuscript carefully.

Response #5:Thank you for your valuable feedback. The authors sincerely apologize for the shortcomings in their English writing and have made comprehensive and thorough revisions throughout the manuscript. These modifications focus on correcting grammatical errors and optimizing sentence structure to ensure clearer and more accurate expression, thereby enhancing the overall academic rigor of the paper.

Reviewer 5 Report

Comments and Suggestions for Authors

Thank you for your response.

Author Response

Thanks.